# Influence of Post-Milking Treatment on Microbial Diversity on the Cow Teat Skin and in Milk

Isabelle Verdier-Metz [1],*, Céline Delbès [1], Matthieu Bouchon [2], Philippe Pradel [2], Sébastien Theil [1], Etienne Rifa [1], Agnès Corbin [3] and Christophe Chassard [1]

1    UMR545 Fromage, INRAE, VetAgro Sup, Université Clermont Auvergne, 20 Côte de Reyne, 15000 Aurillac, France; celine.delbes@inrae.fr (C.D.); sebastien.theil@inrae.fr (S.T.); etienne.rifa@inrae.fr (E.R.); christophe.chassard@inrae.fr (C.C.)

2    UE1414 Herbipôle, INRAE, Université Clermont Auvergne, Domaine de la Borie, 15190 Marcenat, France; matthieu.bouchon@inrae.fr (M.B.); philippe.pradel@inrae.fr (P.P.)

3    Lallemand SAS, 19 Rue des Briquetiers, BP59, 31702 Blagnac, France; acorbin@lallemand.com

*    Correspondence: isabelle.verdier-metz@inrae.fr

**Abstract:** In dairy cattle, teat disinfection at the end of milking is commonly applied to limit colonization of the milk by pathogenic microorganisms via the teat canal. The post-milking products used can irritate the teat skin and unbalance its microbial population. Our study aimed to assess the impact of different milking products on the balance of the microbial communities on the teat skin of cows and in their milk. For 12 weeks at the end of each milking operation, three groups of seven Holstein dairy cows on pasture received either a chlorhexidine gluconate-based product (G) or a hydrocolloidal water-in-oil emulsion (A), or no post-milking product (C). The composition of the bacterial and fungal communities on the teat skin and in the milk were characterized using a culture-dependent method and by high-throughput sequencing of marker genes to obtain amplicon sequence variants (ASVs). The individual microbiota on the cows' teat skin was compared for the first time to that of a cow pool. In contrast to the milk, the post-milking treatment influenced the microbiota of the teat skin, which revealed a high microbial diversity. The water-in-oil emulsion appeared to slightly favour lactic acid bacteria and yeasts and to limit the development of undesirable bacteria such as *Pseudomonas* and *Staphylococcus*.

**Keywords:** dairy cow; post-milking; milk; teat; bacterial community; fungal community; metabarcoding

## 1. Introduction

The origin of microorganisms in raw milk has been the subject of an increasing number of studies over the past decade, many of which have focused on potential sources in the farm environment [1–3]. The composition of milk microbiota depends mostly on that of the microbial ecosystems directly in contact with the milk, such as the teat canal and the teat surface of the udder, and the dairy equipment. In a second phase, it depends on several microbial environmental sources that are not in contact with the milk, such as the bedding material, feces, feeds, drinking and washing water, and the air in the barn, milking parlor, and the milker [4]. The teat skin of cows is located at the crossroads of environmental microbial sources. The teats are considered the major reservoir of the microbial diversity of raw milk since many different bacterial taxa found in raw milk have been identified on the teat skin [5,6]. Doyle et al. (2017) evaluated and classified the possible sources of microbiota for raw milk using high throughput sequencing methods; their study confirmed that the teat skin surface is the most important source, followed by the feces [7]. Some studies have shown that the microbial community on the teat skin of dairy cows varies quantitatively and qualitatively from one farm to another [2,5]. Microorganisms, whether pathogenic or otherwise, can colonize the teat skin through contact with the bedding material, which depends on the animal's feeding and housing conditions. Over the last

few decades, numerous hygienic milking practices have been established (washing of the milking equipment, teat care before and after milking) to reduce or avoid contamination of the milk with pathogens. In addition to eliminating the pathogenic population which is dangerous to humans, such practices can modify the balance of the teat skin microbiota [1,8]. Amongst milking practices, pre-and post-milking teat disinfections are widely used to reduce the microbial population [9,10], limit new intra-mammary infections, and prevent the incidence of mastitis [11]. Post-milking teat disinfection is used immediately after claw piece removal, before the teat canal sphincter begins to close and before any bacteria have the opportunity to colonize and multiply. Post-milking teat disinfection is, however, routinely performed with conventional chemical disinfectants including chlorhexidine, iodine compounds, or quaternary ammonium salts, with variable effectiveness [12] and some resistance [13]. Moreover, chemicals often irritate and dry out the teat skin, which constitutes the first barrier against pathogenic microorganisms. Therefore, post-milking treatments with other chemicals known as emollients and humectants were added to counter these effects and also to soften and improve the teat skin condition after milking. Alternative teat disinfectants have been increasingly researched [14,15], particularly in organic agriculture or in cheese production. Indeed, preserving the integrity of the teat skin microbiota is of potential interest for its role in the health of the animal [16] on the one hand and as a reservoir of microbial diversity for milk and cheese on the other hand [6]. Most studies on post-milking teat disinfectants have focused on their effects on the bacteria responsible for mastitis [11,17,18]. In our study, we investigated the impact of different post-milking teat dips on milk and teat microbiota as a whole. A hydrocolloidal water-in-oil emulsion was compared to a chlorhexidine gluconate-based product and a negative control without a post-milking teat dip. High-throughput sequencing of marker genes to obtain amplicon sequence variants (ASVs) was applied to characterize the composition of bacterial and fungal communities on the teat skin and in the milk of dairy cows receiving a post-milking teat dip or not. We also further evaluated the inter-individual variations in microbial diversity profiles of teat skin over time and in relation to the appearance of the teat-end.

## 2. Materials and Methods

### 2.1. Experimental Design

The animal experiment was conducted at the experimental farm of INRAE in Marcenat (UE1414 Herbipôle, https://doi.org/10.15454/1.5572318050509348E12, accessed on 23 February 2022), located in an upland mountain grassland area of central France. A total of 21 Holstein dairy cows grazing the same paddocks and receiving the same post-milking product G (chlorhexidine gluconate base) and no pre-milking product were divided on May 12 into three equivalent groups (Table S1) according to parity (6 multiparous (3.2 ± 1.13 lactation) and 1 primiparous cow per group), lactation stage (5 cows in early lactation (57.3 ± 13.92 days in milk) and 2 cows in late lactation (219.7 ± 21.08 days in milk)) and level of milk production (28.2 ± 5.71 kg·d$^{-1}$). From June 7 and for 12 weeks, at the end of each milking operation each group of 7 cows received a different post-milking product: (i) group G received product G as previously, (ii) group A received a hydrocolloidal water-in-oil emulsion (Lallemand SAS, France) and (iii) group C did not receive any product (Figure S1).

### 2.2. Sampling

The individual teat skins and the total milk of all the animals in each group were sampled during the evening milking three times at a 3-or 4-day time interval (T1, T2, and T3 = period P1) before June 7 when all the animals received the G product, and then three times in the middle (T4, T5, and T6 = period P2) and at the end (T7, T8, and T9 = period P3) of the 12-week treatment. The individual teat skins were sampled as described by Verdier-Metz et al. (2012), with a single sterile swab (Sodibox) moistened with 10 mL of sterile salt tryptone [5]. Each swab was then placed in an individual

stomacher bag and then stored at 4 °C overnight. The next morning they were blended with 12 mL of salt tryptone (1 g/L) using a stomacher (Bag System, Interscience, Saint-Nom-la-Bretèche, France) and the suspensions were extracted. An identical aliquot of each individual suspension was pooled per group of post-milking treatment. The mixed milk from each group was sampled just after evening milking and immediately stored at 4 °C overnight.

### 2.3. Appearance of Teat-End and Udder Health

The appearance of the teat sphincter was evaluated according to a score chart [19], from 1 to 4 (1: normal sphincter; 2: flexible sphincter; 3: rough sphincter with visible keratinization but inferior to 3 mm; 4: very rough sphincter with visible keratinization superior to 3 mm), at the evening milking twice in P1, and twice at the beginning, in the middle and at the end of the experimental period. At the same time, photos of the teat sphincters were taken and two persons made notes. The SCC (threshold of 300,000 cells) and the California mastitis test (CMT) served to diagnose mastitis along the experiment.

### 2.4. Analyses

The somatic cells of individual milks (SCC) were automatically counted (Fossomatic 5000, Foss System) once a week.

Total mesophilic bacteria, lactic acid bacteria, Gram-negative bacteria, ripening bacteria (Gram-positive catalase-positive bacteria), yeasts, and moulds in individual and pooled teat suspensions and pooled milk samples were enumerated as described by Monsallier et al. (2012).

The individual and pooled teat suspensions were centrifuged (8000× $g$, 20 min, 4 °C) and the supernatants were removed. A total of 200 mL of each milk were centrifuged (5300× $g$, 30 min, 4 °C), then the fat and the supernatant were removed. The pellets obtained from the teat suspensions and milks were mixed with 1 mL sterile PBS (Phosphate-buffered saline), centrifuged (13,000× $g$, 5 min, 4 °C) and stored at −20 °C. Total DNA extraction was performed from the pellets of the individual and pooled teat suspensions and of the milk using a FastDNA Spin Kit for Soil (MP Biomedicals, Eschwege, Germany).

The 16S rRNA genes (1450 bp) of the bacterial population from the teat suspensions and milk samples were pre-amplified using the universal bacterial primers W02 (5′-GNTACCTTGTTACGACTT-3′) and W18 (5′-AGAGTTTGATCMTGGCTCAG-3′), as described by Verdier-Metz et al. (2012), for 17 cycles [5]. The variable region V3–V4 of the 16S rRNA gene (~510 bp) was amplified from 2 μL of pre-amplified DNA using primers MSQ-16SV3F (5′-TACGGRAGGCWGCAG-3) and PCR1R-460 (5′-TTACCAGGGTATCTAATCCT-3), as described by Frétin et al. (2018) [6]. To target the ITS2 region of the fungal population, the extracted DNA from the teat suspensions and milk samples were pre-amplified using the primers NL4 (5′-GGTCCGTGTTTCAAGACGG-3′) and ITS5 (5′-GGAAGTAAAAGTCG TAACAAGG-3′), as described by Irobi et al. (1999), for 17 amplification cycles [20]. Then 2 μL of pre-amplified DNA were amplified with primers ITS3f (5′-GCATCGATGAAGAACG CAGC-3′) and ITS4_KYO1 (5′-TCCTCCGCTTWTTGWTWTGC-3′), as described by Bokulich and Mills (2013), for 30 cycles [21]. All the amplicons were sequenced using Illumina MiSeq technology (INRAE, GeT-PLaGE platform) with the 250 bp paired-end V3 chemistry.

### 2.5. Statistical Analyses and Bioinformatics

Statistical analyses were performed with the software R (version 3.6.3). The scores of the teat sphincters and the levels of somatic cell counts were processed by ANOVA factoring in cow group. The microbial enumerations were processed by ANOVA, factoring in cow group and period. The microbial communities were characterized using the workflow rANOMALY [22], which is mainly based on DADA2 and phyloseq packages and uses amplicon sequence variants (ASV) as taxonomic units. Richness (Chao1) and evenness (Shannon) indexes were used to evaluate the α-diversity whose differences between groups were highlighted using an ANOVA. The differences in community compo-

sition (β-diversity) were estimated with the wUnifrac distance method and a Permanova was carried out to show significant differences between groups. Three methods (DESeq, MetaGenomeSeq and MetaCoder) were performed to assess differentially abundant taxa between conditions.

## 3. Results

### 3.1. Teat Condition and Somatic Cell Counts

The teat appearance scores (Table 1) were similar from one group to another when the animals all received the G post-milking teat dip during the P1 period. Gradually, group C reached the best score with 1.13 at the end of the experiment, while group G presented the worst one with 2.17. An intermediate score of 1.56 was obtained by group A. Note 4, corresponding to hyperkeratosis, was not ascribed to any animal, whatever the group considered. In addition, we observed that the sphincter condition of cows C was improved over time, unlike those of cows G, while those of cows A were unchanged.

**Table 1.** Average scores for the appearance of sphincters.

|  | A | G | C | SEM | *p*-Value |
|---|---|---|---|---|---|
| Before experiment (P1) | 1.38 | 1.64 | 1.50 | 0.093 | 0.541 |
| Beginning of experiment (P2) | 1.62 [a,b] | 1.73 [b] | 1.25 [a] | 0.067 | 0.006 |
| Middle of experiment | 1.64 [b] | 2.00 [b] | 1.07 [a] | 0.077 | <0.001 |
| End of experiment (P3) | 1.56 [b] | 2.17 [c] | 1.13 [a] | 0.077 | <0.001 |

[a–c] Means within a row with differing superscript letters differ ($p < 0.05$).

No clinical mastitis and only sporadic subclinical mastitis (one in each group) were observed. The SCC of the three groups varied on average between 30,000 and 600,000 cells/mL with a majority less than 300,000 cells/mL. The SCC in group A was slightly lower than in those of the other two groups except at the end of the experiment where it was not significantly different from those of the G and C groups (Table 2).

**Table 2.** Somatic cell counts.

| Log(SCC/1000) | A | G | C | SEM | *p*-Value |
|---|---|---|---|---|---|
| Before experiment (P1) | 1.66 | 1.99 | 1.83 | 0.064 | 0.081 |
| Beginning of experiment (P2) | 1.57 [a] | 1.88 [b] | 1.80 [a,b] | 0.043 | 0.008 |
| Middle of experiment | 1.84 [a] | 1.91 [a] | 2.31 [b] | 0.058 | 0.001 |
| End of experiment (P3) | 2.00 | 2.00 | 2.25 | 0.053 | 0.084 |

[a,b] Means within a row with differing superscript letters differ ($p < 0.05$).

### 3.2. Microbial Enumeration Using Culture Methods

The concentrations of the different microbial groups were evaluated through a cultural approach, in the milk as well as on the teat surface (Table 3). Concentrations of total mesophilic bacteria in milk ranged between 3.5 and 3.9 $\log_{10}$(cfu/mL). The counts of lactic acid bacteria and of yeasts ranging between 2.3 and 3 $\log_{10}$(cfu/mL) differed significantly ($p < 0.05$ and $p < 0.001$ respectively) according to the post-milking treatment: their concentrations in milk A were the highest while those in milk G were the lowest. The treatment period significantly ($p < 0.01$) affected the counts of lactic acid bacteria, ripening population, yeasts, and moulds, which increased from 0.5 to 1 $\log_{10}$(cfu/mL) between P1 and both the following periods irrespective of the post-dipping group, as was also observed for the SCC. On the teat surface, presumed ripening bacteria and Gram-negative bacteria represented the major bacterial populations, followed by lactic acid bacteria. The microbial concentrations from the individual teat suspensions were slightly lower than those from the pooled ones. Their differences according to the group were similar and not significant, with the exception of Gram-negative bacteria which were lower in number in group A compared to the other two groups ($p < 0.05$). As in the milk, the lactic acid bacteria count was slightly

higher on teat skin A than on G or C and all the microbial levels in P1 were significantly ($p < 0.001$) lower than those of the two other periods. Apart from the milks, the effect of the period on the microbial counts was greater than that of the group (Figure S2).

**Table 3.** Effect of post-milking treatment and period on milk and teat microbial concentrations.

| Microbial Populations | Group [1] | | | Period [2] | | | | *p*-Value | | |
|---|---|---|---|---|---|---|---|---|---|---|
| | A | G | C | P1 | P2 | P3 | SEM | Group | Period | Group × Period |
| Milk [3] (*n* = 27) | | | | | | | | | | |
| Total mesophilic bacteria | 3.55 | 3.87 | 3.80 | 3.90 | 3.57 | 3.75 | 0.19 | 0.24 | 0.30 | 0.12 |
| Gram-negative bacteria | 2.73 | 2.50 | 2.66 | 2.36 | 2.80 | 2.74 | 0.20 | 0.50 | 0.08 | 0.98 |
| Lactic acid bacteria | 2.88 [a] | 2.35 [b] | 2.57 [a,b] | 2.22 [b] | 2.93 [a] | 2.65 [a,b] | 0.19 | 0.05 | 0.005 | 0.54 |
| Ripening population | 2.68 | 2.41 | 2.47 | 1.93 [b] | 2.96 [a] | 2.66 [a] | 0.12 | 0.11 | <0.001 | 0.27 |
| Yeasts | 2.99 [a] | 2.42 [b] | 2.65 [b] | 2.41 [b] | 2.77 [a] | 2.88 [a] | 0.10 | <0.001 | 0.001 | 0.03 |
| Moulds | 0.77 | 0.22 | 0.64 | 0.16 [b] | 1.06 [a] | 0.42 [b] | 0.23 | 0.07 | <0.001 | 0.20 |
| Pooled teat suspensions [4] (*n* = 27) | | | | | | | | | | |
| Total mesophilic bacteria | 4.92 | 5.09 | 5.26 | 4.16 [c] | 5.88 [a] | 5.23 [b] | 0.19 | 0.22 | <0.001 | 0.59 |
| Gram-negative bacteria | 3.87 | 4.14 | 4.37 | 2.56 [b] | 4.93 [a] | 4.90 [a] | 0.28 | 0.23 | <0.001 | 0.79 |
| Lactic acid bacteria | 3.14 | 2.50 | 2.64 | 2.15 [b] | 3.47 [a] | 2.65 [a,b] | 0.38 | 0.24 | 0.01 | 0.09 |
| Ripening population | 4.13 | 4.34 | 4.08 | 3.27 [b] | 4.73 [a] | 4.54 [a] | 0.39 | 0.77 | <0.001 | 0.15 |
| Yeasts | 2.12 | 2.07 | 2.13 | 1.67 [b] | 2.22 [a,b] | 2.45 [a] | | 0.97 | 0.01 | 0.77 |
| Moulds | 1.34 | 1.55 | 1.64 | 1.35 | 1.82 | 1.40 | | 0.44 | 0.13 | 0.55 |
| Individual teat suspensions [4] (*n* = 189) | | | | | | | | | | |
| Total mesophilic bacteria | 4.77 | 4.81 | 4.91 | 3.66 [c] | 5.71 [a] | 5.11 [b] | 0.11 | 0.43 | <0.001 | 0.03 |
| Gram-negative bacteria | 3.38 [b] | 3.70 [a,b] | 3.78 [a] | 1.78 [b] | 4.63 [a] | 4.44 [a] | 0.17 | 0.04 | <0.001 | 0.44 |
| Lactic acid bacteria | 2.72 | 2.48 | 2.51 | 1.64 [b] | 3.30 [a] | 2.77 [a] | 0.17 | 0.30 | <0.001 | 0.02 |
| Ripening population | 4.00 | 3.96 | 4.19 | 3.13 [c] | 4.68 [a] | 4.33 [b] | 0.11 | 0.08 | <0.001 | <0.001 |
| Yeasts | 1.71 | 1.72 | 1.77 | 1.14 [b] | 1.93 [a] | 2.14 [a] | | 0.86 | <0.001 | 0.57 |
| Moulds | 1.34 | 1.33 | 1.52 | 1.07 [c] | 1.71 [a] | 1.40 [b] | | 0.12 | <0.001 | 0.34 |

[a–c] Means within a row with differing superscript letter differ ($p < 0.05$). [1] Group of post-milking treatment: A = hydrocolloidal water-in-oil emulsion; G = chlorhexidine gluconate based; C = no post-milking; [2] Period of sampling; [3] Milk concentrations are expressed in Log (cfu/mL of milk); [4] Teat concentrations are expressed in Log (cfu/mL of teat suspensions).

### 3.3. Microbial Diversity Assessment Using High Throughput Sequencing Methods

The microbial profiles of the milks and the pooled and individual teat suspensions were obtained with an HTS approach based on Amplicon Sequence Variants (ASV) targeting 16S rRNA and ITS2 genes. Alpha diversity, representing the microbial variation inside a single sample, was estimated through a richness estimator index (Chao1) and a richness-evenness estimator index (Shannon). No impact of treatment was observed on the alpha-diversity indexes from any samples of milk, whether individual or pooled teat samples, or for bacteria as fungi (Table 4). The Chao1 index for bacteria in milk A increased progressively and significantly throughout the experiment, while it increased then decreased for milks G and C. Regardless of the post-milking treatment, the period had a significant effect on the bacterial and fungal alpha-diversity indexes of the individual teat suspensions, contrary to the pooled ones.

**Table 4.** Alphadiversity indexes.

| | **Group** | | | | **Period** | | | | **Group × Period** [1] |
|---|---|---|---|---|---|---|---|---|---|
| | **A** | **G** | **C** | ***p*-Value** | **P1** | **P2** | **P3** | ***p*-Value** | ***p*-Value** |
| 16S (Bacteria) | | | | | | | | | |
| Chao1 | | | | | | | | | |
| Milk | 79.1 | 90.0 | 98.2 | 0.66 | 65.1 [b] | 114.2 [a] | 82.9 [b] | 0.003 | 0.02 |
| Individual Teat | 110.2 | 115.8 | 109.0 | 0.55 | 118.9 [a] | 124.9 [a] | 92.3 [b] | <0.001 | <0.001 |
| Pooled Teat | 130.5 | 116.9 | 100.6 | 0.11 | 134.6 | 120.2 | 85.0 | 0.05 | 0.05 |
| Shannon | | | | | | | | | |
| Milk | 3.1 | 3.4 | 3.4 | 0.16 | 3.2 | 3.5 | 3.2 | 0.09 | 0.20 |
| Individual Teat | 3.0 | 3.1 | 3.0 | 0.51 | 3.0 [ab] | 3.1 [a] | 2.9 [b] | 0.02 | 0.002 |
| Pooled Teat | 3.3 | 3.2 | 3.1 | 0.55 | 3.2 | 3.3 | 3.2 | 0.53 | 0.40 |
| ITS2 (Fungi) | | | | | | | | | |
| Chao1 | | | | | | | | | |
| Milk | 15.9 | 19.6 | 15.7 | 0.09 | 14.3 | 20.8 | 15.6 | 0.06 | 0.07 |
| Individual Teat | 43.9 | 43.3 | 43.2 | 0.99 | 27.9 [c] | 60.6 [a] | 43.8 [b] | <0.001 | <0.001 |
| Pooled Teat | 58.0 | 51.5 | 25.6 | 0.20 | 31.5 | 68.4 | 26.5 | 0.06 | 0.21 |
| Shannon | | | | | | | | | |
| Milk | 1.2 | 1.2 | 1.3 | 0.61 | 1.3 | 1.3 | 1.2 | 0.46 | 0.35 |
| Individual Teat | 2.4 | 2.6 | 2.6 | 0.36 | 1.9 [b] | 3.1 [a] | 2.6 [a] | 0.0010 | 0.02 |
| Pooled Teat | 2.6 | 2.4 | 2.0 | 0.55 | 2.0 | 3.0 | 1.7 | 0.14 | 0.48 |

[a–c] Means within a row with differing superscripts differ according to the Tukey difference test; [1] Pairwise interaction between Group and Period.

Beta diversity, representing the variation in microbial communities composition between samples was evaluated by a Multidimensional scaling (MDS) plot based on the weighted Unifrac (wUF) distance matrix. In our study, no significant difference in beta diversity was observed between the microbiota of milks, regardless of the post-milking teat treatment. According to the beta-diversity analysis, the microbiota in the pooled teat suspensions did not differ significantly from that of the individual teat suspensions (Figure S3), except in the period P3 for the fungi. The beta diversity between pooled teat suspensions was not significantly different according to the post-milking teat dip (Table S2), unlike that of the individual teat suspensions (Figure 1a) for which the bacterial diversity of teat suspensions G differed ($p = 0.002$) from that of the A and C for the period P3. The beta diversity of individual teat suspensions showed significant differences ($p < 0.05$) between the three periods for each treatment, for both 16S and ITS ASVs (Figure 1b). Concerning bacterial profiles, the beta diversity index in period P3 differed ($p = 0.001$) from that of the other periods for samples G and C, while the A beta diversity seemed to be more stable over time. For the fungal profiles, the beta diversity index showed large differences ($p = 0.001$) between P1 and the other two periods. This period-dependent differentiation of fungal communities was coupled with a decreasing inter-individual variability at periods P2 and P3 compared to P1, irrespective of the post-dipping group (Figure 1a; Table S3).

### 3.4. Phylogenetic Profiles of Teat and Milk Microbial Communities

A high-resolution method based on ASV inference was used to identify 478 bacterial ASVs and 94 fungal ASVs shared by the milks and the pooled and individual teat suspensions (Figure S4), of which 26 and 35 respectively each represented more than 1% of the total sequence reads. The 585 bacterial and 108 fungal ASVs identified in all the milks were assigned to 146 and 60 genera, respectively. Seventeen bacterial ASVs, each present at more than 1%, accounted in total for 60.8% of the total sequences (Table 5) and were assigned to 13 genera.

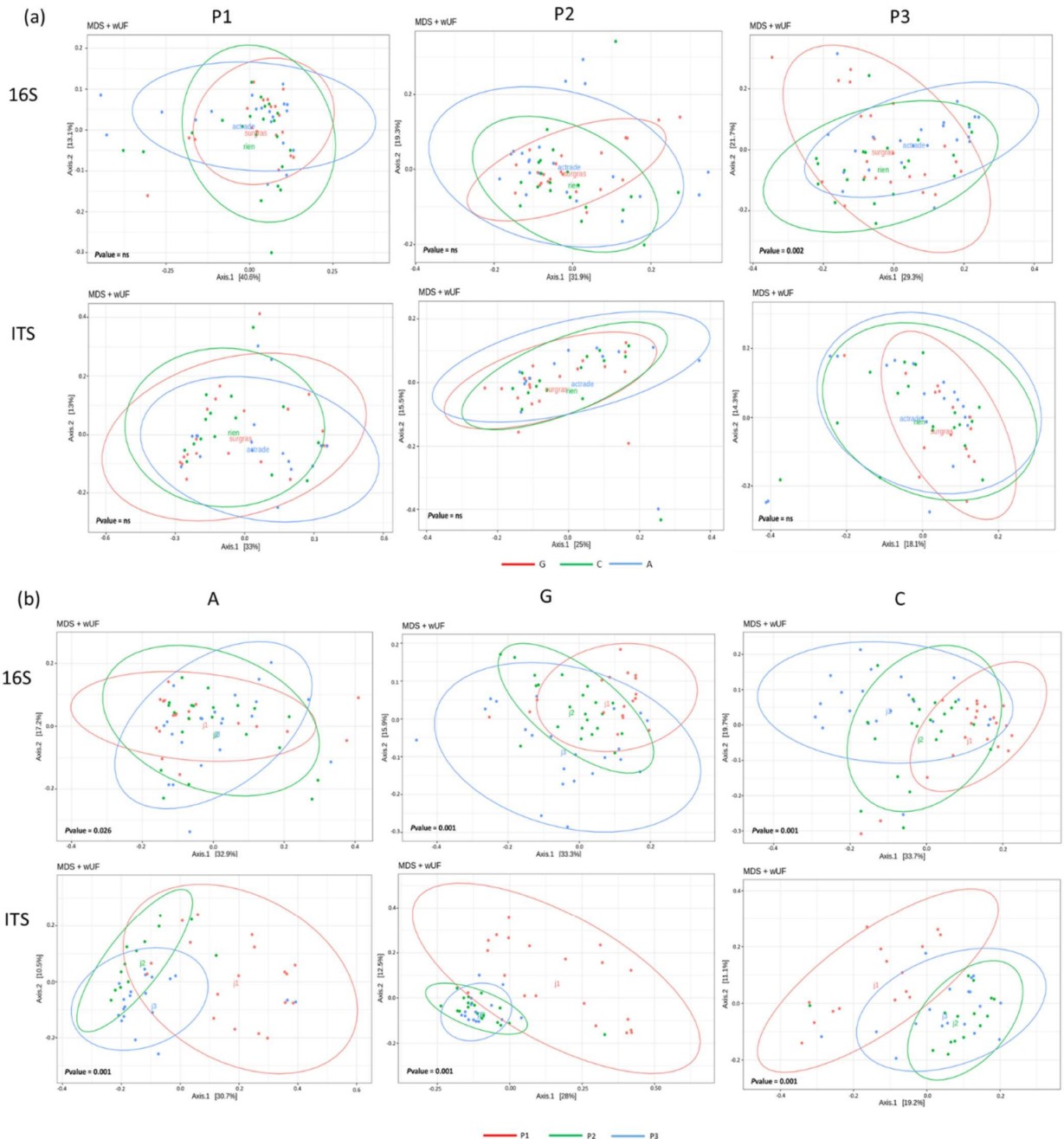

**Figure 1.** Bacterial (16S) and fungal (ITS) betadiversity of individual teat suspensions according to (**a**) the treatment at each period and (**b**) the period for each treatment.

**Table 5.** Bacterial and fungal ASVs with mean abundance above 1% of the total sequences in milks and in the pooled and individual teat suspensions.

| | | Abundance (%) | | |
|---|---|---|---|---|
| | | | Teat | |
| ASV | Species | Milk | Pool | Individual |
| Bacteria (16S rRNA) | *Actinobacteria* | | | |
| 46fe2c89eaf45507201f73d05f7dd682 | *Kocuria salsicia* | 1.00 | – | 0.00 |
| 4c0a7b78bc28cd297d940358c90fca11 | *Kocuria salsicia_varians* | 1.92 | 0.08 | 0.09 |
| | *Bacteroidetes* | | | |
| 45a0f682d1b59ec4c012c6edc5a8593f | *Chryseobacterium haifense* | 1.05 | 0.02 | 0.05 |
| | *Firmicutes* | | | |
| 88a4abe161399fb663f20a3249edd2c6 | *Aerococcus urinaeequi_viridans* | 1.67 | 3.58 | 3.48 |
| bdcf0a98c3a746ae82f10262eb2cc273 | *Clostridioides difficile_bartlettii* | 2.62 | 6.11 | 6.08 |
| 260ed04a6ad8d8fb320e4803f51ee938 | *Clostridium disporicum_saudiense* | 0.76 | 1.65 | 1.66 |
| 662e2046cdeb3803a324da7c7d4b55ed | *Clostridium disporicum_saudiense* | 0.28 | 1.14 | 1.04 |
| 4dbaad38558ae942a1e56ae96ba14f4a | *Exiguobacterium aestuarii* | 0.04 | 1.59 | 2.65 |
| 2535950da4f5ce2e05509946880e1912 | *Lactococcus lactis* | 7.05 | 0.05 | 0.07 |
| 19c50dacd1bcf20c7ee29330b0565eb6 | *Pediococcus stilesii_acidilactici* | 1.36 | 0.00 | 0.01 |
| 851262e388f9eda58b0f5016acc737fe | *Pediococcus stilesii_acidilactici* | 1.03 | 0.00 | 0.01 |
| de38e8e17d9c8f260f6dbb99561881ab | *Paeniclostridium sordellii* | 4.99 | 10.40 | 9.92 |
| e0c59b317f7537c66626d5f49ea44a89 | *Romboutsia sedimentorum* | 1.63 | 3.19 | 3.45 |
| ff2c64ded51bdfc792648dca80a3d375 | *Romboutsia timonensis* | 11.13 | 23.12 | 21.65 |
| 09e64bf0e07c9a7959b90f5bb78d63b2 | *Staphylococcus aureus* | 1.09 | 0.21 | 0.26 |
| 3ea7d63cf8427d340ff7a71adf338fee | *Staphylococcus haemolyticus_petrasii* | 0.29 | 1.30 | 1.73 |
| 42697dcb9c518dcc6bf2e48c1ae1b276 | *Staphylococcus petrasii* | 6.13 | 2.73 | 3.07 |
| ad306fc9d2bd79082dc8e044dfe8588d | *Streptococcus uberis_porcinus* | 1.01 | 0.01 | 0.02 |
| 56deff739fb6bfbc421cccbba6170ad6 | *Turicibacter sanguinis* | 2.14 | 5.06 | 4.52 |
| | *Proteobacteria* | | | |
| 2d711873254cf9212d36b48150030c42 | *Acinetobacter albensis_lwoffii* | 0.01 | 1.08 | 0.58 |
| 239d5c60225d2e64c348234e9a85de3b | *Acinetobacter haemolyticus* | 0.01 | 1.65 | 1.43 |
| b7176533054c4a045208143afd50f67b | *Acinetobacter indicus* | 0.32 | 6.01 | 3.23 |
| 3239a6358dec42f7d5288124e6639ebd | *Moraxella osloensis* | 5.42 | 0.64 | 1.87 |
| 2e127c4643317603c1522a72a25ab663 | *Pseudomonas al-caliphila_chengduensis_toyotomiensis_oleovorans* | | 1.02 | 1.57 |
| 542870b5c028dea2cf7adbddcf28f104 | *Pseudomonas indoloxydans* | 0.08 | 4.37 | 6.17 |
| f2a166e1c7a8de0a617183907f36e3ff | *Ralstonia pickettii* | 9.52 | 0.22 | 0.44 |
| Fungi (ITS) | Moulds | | | |
| be901ea53977fe923f96a7495b0bd739 | *Ampelomyces quisqualis* | 0.00 | 1.16 | 0.03 |
| daf19ea76f64fac5bdbd4ee6db820021 | *Cladosporium crousii_pini-ponderosae_colombiae* | 0.04 | 3.68 | 4.41 |
| 619fa4027595729b90c2b18fb38ff24a | *Cladosporium subcinereum_antarcticum_phlei_macrocarpum* | 0.03 | 3.02 | 2.22 |
| 17f87672bb942ca8b3596354625b1542 | *Claviceps humidiphila* | 0.01 | 1.48 | 1.89 |
| aeb121cc9a64ee4fe15af0b356649b27 | *Claviceps macroura* | 0.00 | 2.66 | 0.96 |
| 714228661ee55d498566562d4d782b62 | *Claviceps macroura* | - | 1.89 | 1.02 |
| 6c95e6c5792d5e1af4b184f43b6eb6b6 | *Claviceps pazoutovae_monticola* | 0.00 | 1.34 | 0.83 |
| 67602484504bca4f1a2105f6c222cf0d | *Epicoccum phragmospora_Nothophoma macrospora* | 0.05 | 2.10 | 2.82 |
| 5ca556568551faf004e83e42c1d1c765 | *Neoascochyta cylindrispora_desmazieri* | 0.01 | 0.51 | 1.26 |
| 38ab9a64ce7271afcf3101ac717dc661 | *Neosetophoma phragmitis* | 0.02 | 2.11 | 3.07 |
| 0a8cc784caae3436b2a2f56df6cec4c0 | *Neosetophoma phragmitis* | 0.00 | 1.67 | 3.33 |
| 5663634563309440e3ca3696c23088f6 | *Penicillium fuscoglaucum_caseifulvum_commune* | - | 1.28 | 0.00 |
| b672393eff9c6c00f455ea2f5a3acfcf | *Preussia persica* | 0.01 | 1.22 | 0.82 |
| 3fb396dc4a7b06fbd95897929a06723a | *Pseudoconiothyrium broussonetiae* | 0.00 | 0.73 | 1.11 |
| a4234fd305f00f140744be4bb2e543c1 | *Pseudopithomyces rosae* | 0.03 | 1.39 | 1.99 |
| 63baa23ebd09803e4a135175228373e6 | *Pyrenochaetopsis microspora_leptospora* | - | 0.85 | 1.03 |
| 5c6b15eadc471efbba1b5044b740272c | *Thelebolus spongiae_ellipsoideus* | 0.00 | 1.97 | 2.26 |
| 734cc9d4e22dae3261bac2e7a68c972e | *Ustilago nunavutica* | 0.00 | 1.27 | 2.74 |

**Table 5.** *Cont.*

| ASV | Species | Abundance (%) | | |
| --- | --- | --- | --- | --- |
| | | | Teat | |
| | | Milk | Pool | Individual |
| 82a81b8fe817daccea54156e773630ac | *Ustilago nunavutica* | 0.01 | 0.58 | 1.42 |
| 5beac8cb91c1c4f4e0a45c78e5e9d0af | *Vacuiphoma oculihominis_Neodidymella thailandicum* | - | 1.31 | 2.20 |
| cf14789892a27b7ecafb127f44bc2130 | *Vishniacozyma victoriae* | 0.00 | 0.79 | 1.22 |
| fcd5aded7ad3e8947a2ac66434368f80 | *Xenopyrenochaetonopsis* | - | 1.16 | 0.12 |
| | Yeasts | | | |
| 66d5d96e85ffe8985a9a7957c8b6d932 | *Candida inconspicua* | 28.97 | 0.19 | 0.59 |
| d51af0a89a56f6289e89e6858ca54857 | *Candida inconspicua _Pichia cactophila* | 32.61 | 0.62 | 0.81 |
| b8672e965ae0ef9638e7424195de16d5 | *Candida pseudoglaebosa* | 3.11 | 0.11 | 0.04 |
| ddc456b20cd74e9247f05913562029b0 | *Candida santamariae var. membranifaciens* | 3.42 | 5.23 | 1.81 |
| 2ffbdc8725b87e1923c46eeb2e9e8c50 | *Cutaneotrichosporon curvatum* | 0.09 | 2.72 | 2.28 |
| c8a80e78d59949c51837339037dad7a3 | *Cutaneotriphosporon curvatum_cyanovorans* | 0.10 | 0.42 | 1.10 |
| 53193ba9d7fda4ce594b108dae39bc83 | *Debaryomyces prosopidis_vindobonensis_fabryi* | - | 7.52 | 0.07 |
| 0a5c97aafaa956e98ed3b0984ed1c8b2 | *Geotrichum silvicola* | 26.63 | 3.72 | 3.04 |
| 47597a8681f2304dd2d2a753350eb50a | *Kluyveromyces lactis* | 1.36 | 0.57 | 1.76 |
| 5feaa83003fb3b799ee8ffa513d6f08c | *Malassezia restricta* | 0.00 | 1.54 | 0.01 |
| 6e4adb61e7eeda079bf2af3dd2806138 | *Sporobolomyces ruberrimus* | 0.00 | 1.19 | 1.04 |
| 42dc2fa2227e97ab47f9619dbfff3159 | *Trichosporon aquatile* | 0.12 | 0.57 | 1.11 |
| cc81bbf3a7e847b76b28c7359999852b | *Wickerhamiella shivajii* | 0.85 | 19.18 | 14.10 |

Highest value (32.61) ▬▬▬▬▬▬▬ (0.00) Lowest value.

Eight of them belonged to the *Firmicutes* phylum: *Aerococcus* (1ASV), *Clostridioides* (1 ASV), *Lactococcus* (1 ASV), *Pediococcus* (2ASV), *Romboutsia* (2ASV), *Staphylococcus* (2 ASV), *Streptococcus* (1 ASV) and *Turicibacter* (1 ASV). Six fungal ASVs present at more than 1% were yeasts and accounted for 96.1% of the total fungi sequences: four ASVs were assigned to the genus *Candida* (68.1%), one to *Geotrichum* (26.6%), and one to *Kluyveromyces* (1.4%). Sixteen of the 648 bacterial ASVs identified in all pooled teat suspensions and 16 of the 795 identified in all individual teat suspensions were present at more than 1% abundance. They accounted for 74% and 73.5% of the sequences, respectively: the abundance of each ASV was quite similar between the pooled and individual suspensions. Fifteen ASVs common to both the pooled and individual samples (Table 4) were assigned to 11 genera, of which six were also present above 1% in milk, namely *Aerococcus* (one ASV), *Clostridioides* (one ASV), *Moraxella* (one ASV), *Paeniclostridium* (one ASV), *Romboustia* (two ASVs) and *Turicibacter* (one ASV). The five other genera, *Acinetobacter* (three ASVs), *Clostridium* (two ASVs), *Exiguobacterium* (one ASV), *Pseudomonas* (two ASVs), and *Staphylococcus* (two ASVs), were absent or present at less than 1% in the milk. We also identified 317 and 439 fungal ASVs in all of the pooled and individual teat suspensions, respectively. Both counted 24 fungal ASVs present at more than 1% for a total abundance of 71.2% and 72.8%, respectively (Table 4), of which 16 were shared between the pooled and individual suspensions. Apart from *Wickerhamiella shivajii*, never found in the dairy system, which was the most abundant yeast in both teat suspension types, the pooled teat suspensions distinguished themselves from the individual samples by the prevalence of two yeasts (*Debaryomyces* and *Candida*).

The combination of three differential analysis methods for the abundance of bacterial and fungal ASVs highlighted abundance differences between the three post-milking treatments. A maximum of two of the three methods used revealed differences in the abundance of the ASVs present at more than 1% of the total abundance between the three treatments according to the period (Table 6). At the end of the experiment (P3), bacterial taxa were not differentially abundant in the milks between the three treatments, while a higher proportion of *Kluyveromyces lactis* and *Candida santamariae* differentiated milks G from milks A. In period P1, *Staphylococcus aureus* was less abundant in milks A than in the

other two, whereas *Romboutsia timonensis* was more abundant in milks G and *Kluyveromyces lactis* in milks C. In period P2, milks A were distinguished from G and C by a greater abundance of four bacterial genera assigned to *Kocuria*, *Moraxella*, *Pediococcus*, and *Ralstonia*, together with several ASVs assigned to the genus, *Candida*. In period P3, several microbial and fungal taxa differentiated the teat skin suspensions between the three post-milking treatments: four bacterial ASVs assigned to *Clostridioides*, *Plaeniclostridium*, and *Romboutsia* and eight fungal ASV (three yeasts and five moulds) dominant on teat skin A. *Exiguobacterium*, *Acinetobacter* and *Clostridioides* were the most abundant taxa identified in teat suspensions C, while *Pseudomonas*, *Staphylococcus* and *Penicillium* dominated on teat skin C. A differential analysis of the microbial taxa in the teat suspensions showed that the number of differentially abundant taxa between the three treatments was higher in the individual teat suspensions for bacteria and in the pooled samples for fungi.

**Table 6.** Differential analysis of bacterial (a) and fungal (b) ASVs present at more than 1% of abundance in milks and in the pooled and individual teat suspensions according to the period.

| (a) Bacterial Taxa | Period | Treatments Compared | Statistical Methods [1] | Treatment with Highest abundance [2] |
|---|---|---|---|---|
| Milk | | | | |
| *Staphylococcus aureus* | P1 | A_vs._G | 1 | G |
| *Romboutsia timonensis* | P1 | C_vs._G | 1 | G |
| *Staphylococcus aureus* | P1 | A_vs._C | 1 | C |
| *Pediococcus stilesii_acidilactici* | P2 | A_vs._G | 1,2 | A |
| *Pediococcus stilesii_acidilactici* | P2 | A_vs._G | 2 | A |
| *Moraxella osloensis* | P2 | A_vs._C | 1 | A |
| *Kocuria salsicia* | P2 | A_vs._C | 1 | A |
| *Kocuria salsicia_varians* | P2 | A_vs._C | 1 | A |
| *Ralstonia pickettii* | P2 | A_vs._C | 1,3 | C |
| Pooled teat suspension | | | | |
| *Staphylococcus petrasii* | P1 | C_vs._G | 1 | G |
| *Turicibacter sanguinis* | P1 | C_vs._G | 1 | C |
| *Staphylococcus petrasii* | P3 | C_vs._G | 1 | C |
| *Clostridioides difficile_bartlettii* | P3 | A_vs._C | 1 | A |
| Individual teat suspension | | | | |
| *Acinetobacter haemolyticus* | P1 | A_vs._G | 1 | G |
| *Acinetobacter indicus* | P1 | A_vs._G | 1,3 | G |
| *Acinetobacter haemolyticus* | P1 | A_vs._C | 2 | C |
| *Acinetobacter indicus* | P1 | A_vs._C | 2,3 | C |
| *Exiguobacterium aestuarii* | P2 | A_vs._G | 3 | G |
| *Paeniclostridium sordellii* | P2 | C_vs._G | 3 | G |
| *Romboutsia sedimentorum* | P2 | C_vs._G | 3 | G |
| *Romboutsia timonensis* | P2 | C_vs._G | 3 | G |
| *Clostridioides difficile_bartlettii* | P3 | A_vs._C | 3 | A |
| *Plaeniclostridium sordellii* | P3 | A_vs._C | 3 | A |
| *Romboustia sedimentorum* | P3 | A_vs._C | 3 | A |
| *Romboustia timonensis* | P3 | A_vs._C | 3 | A |
| *Exiguobacterium aestuarii* | P3 | A_vs._G | 1,3 | G |
| *Acinetobacter indicus* | P3 | C_vs._G | 1 | G |
| *Clostridioides difficile_bartlettii* | P3 | C_vs._G | 1,3 | G |
| *Pseudomonas al-caliphila_chengduensis_toyotomiensis_oleovorans* | P3 | A_vs._C | 3 | C |
| *Pseudomonas al-caliphila_chengduensis_toyotomiensis_oleovorans* | P3 | C_vs._G | 3 | C |
| *Pseudomonas indoloxydans* | P3 | C_vs._G | 3 | C |
| *Staphylococcus haemolyticus_petrasii* | P3 | C_vs._G | 3 | C |
| *Staphylococcus petrasii* | P3 | C_vs._G | 3 | C |

**Table 6.** *Cont.*

| (b) Fungal Taxa | Period | Treatments Compared | Statistical Methods [1] | Treatment with Highest Abundance [2] |
|---|---|---|---|---|
| Milk | | | | |
| *Candida inconspicua* | P1 | A_vs._C | 1 | A |
| *Candida inconspicua _Pichia cactophila* | P1 | A_vs._C | 1 | A |
| *Kluyveromyces lactis* | P1 | A_vs._C | 1 | C |
| *Kluyveromyces lactis* | P1 | C_vs._G | 1 | C |
| *Candida inconspicua* | P2 | A_vs._G | 1 | A |
| *Candida inconspicua _Pichia cactophila* | P2 | A_vs._G | 1 | A |
| *Candida inconspicua* | P2 | A_vs._C | 1 | A |
| *Candida inconspicua _Pichia cactophila* | P2 | A_vs._C | 1 | A |
| *Candida santamariae var. membranifaciens* | P2 | A_vs._G | 1 | G |
| *Candida santamariae var. membranifaciens* | P2 | A_vs._C | 1 | C |
| *Kluyveromyces lactis* | P3 | A_vs._G | 1 | G |
| *Candida santamariae var. membranifaciens* | P3 | A_vs._G | 1 | G |
| Pooled teat suspension | | | | |
| *Xenopyrenochaetonopsis 89%* | P2 | A_vs._G | 1 | A |
| *Ampelomyces quisqualis* | P2 | A_vs._C | 1 | C |
| *Ampelomyces quisqualis* | P2 | C_vs._G | 1 | C |
| *Xenopyrenochaetonopsis 89%* | P2 | C_vs._G | 1 | C |
| *Cutaneotrichosporon curvatum* | P3 | A_vs._C | 1 | A |
| *Neosetophoma phragmitis* | P3 | A_vs._C | 1 | A |
| *Debaryomyces prosopidis_vindobonensis_fabryi* | P3 | A_vs._C | 1 | A |
| *Vacuiphoma oculihominis_Neodidymella thailandicum* | P3 | A_vs._C | 1 | A |
| *Epicoccum phragmospora_Nothophoma macrospora_Verrucoconiothyrium eucalyptigenum* | P3 | A_vs._C | 1 | A |
| *Sporobolomyces ruberrimus* | P3 | A_vs._C | 1 | A |
| *Pseudopithomyces rosae* | P3 | A_vs._C | 1 | A |
| *Cladosporium crousii_pini-ponderosae_colombiae* | P3 | A_vs._C | 1 | A |
| *Penicillium fuscoglaucum_caseifulvum_commune* | P3 | A_vs._C | 1 | C |
| Individual teat suspension | | | | |
| *Wickerhamiella shivajii* | P1 | A_vs._C | 3 | A |
| *Claviceps humidiphila* | P1 | A_vs._G | 1 | G |
| *Kluyveromyces lactis* | P1 | C_vs._G | 1,3 | G |
| *Candida santamariae var. membranifaciens* | P1 | C_vs._G | 1 | G |
| *Geotrichum silvicola* | P1 | A_vs._C | 3 | C |
| *Claviceps humidiphila* | P1 | A_vs._C | 1,3 | C |
| *Geotrichum silvicola* | P3 | C_vs._G | 1 | C |

[1] Methods for which ASV was differentially abundant between treatments: 1 = DESeq; 2 = MetaGenomSeq; 3 = MetaCoder; [2] Treatment among the paired treatments compared, for which the considered ASV was significantly more abundant.

## 4. Discussion

Many studies have compared teat dips after milking and their relevance to clinical or sub-clinical mastitis, milk contamination, or teat condition [23–25]. Some of them aimed at comparing different teat dipping products [10,26], but a few studies [27,28] have compared post-milking teat dipping products to a control without teat dipping, as in our work. However, to our knowledge, no study has compared data from individual teat sampling of cows with data from a mix of the same animals using the HTS approach at an amplicon sequence variant (ASV) level.

Commonly practiced on dairy farms, teat dipping at the end of milking has a twofold objective: (i) a bactericidal action against germs that would take advantage of the opening of the sphincters to colonize the udder quarters, and (ii) the preservation or improvement

of the hydration and elasticity of the teat skin by using emollients or softeners. Under the summer conditions of our study, none of the three treatments tested had any impact either on the occurrence of mastitis (one sub-clinical mastitis in each group and no clinical mastitis) or on teat-end hyperkeratosis, which is known as a risk factor for clinical or subclinical mastitis [29]. As reported by Morrill et al. (2019) with a powder chlorhexidine acetate-based post-milking treatment [26], our product G, which was gluconate-based in liquid form, led to a lower teat end score compared to treatments C and A. It is also generally accepted that, apart from obvious health problems, the cessation of summer dipping for grazing cows has no effect on teat health. Indeed, group C, receiving no postmilking product, even saw its average score improve. Post-milking product A seemed to have maintained the state of the sphincter.

The levels of total mesophilic bacteria in milk were consistent with those previously observed on several farms [2,30]. The levels of lactic acid bacteria and yeasts were significantly higher in milk A than in milk G and the levels in milk C were intermediate. In a previous study comparing an iodine product to no post-dipping [31], the microbial counts of the milk were not affected by the post-milking treatment. Widely studied and referenced in the literature, the essential role of lactic acid bacteria and yeasts in the manufacturing process of dairy products is no longer open to question [4,32,33]. The identification of nearly 700 microbial ASVs in the milk samples confirmed their high taxonomic complexity [34–36], which was slightly impacted by the post-milking treatment, in line with Doyle et al. (2017) who showed that the habitat of the cows (outdoor vs indoor) had a greater impact on raw milk microbiota than teat preparation [1]. Only a higher abundance of *Kluyveromyces lactis* and *Candida* was observed in C compared to A at the end of the experiment. Compared to the bacterial community, fungal diversity in milk is relatively unknown and few works have dealt with ITS gene amplicon sequencing. The yeasts *Candida* and *Kluyveromyces lactis* identified in our milks have been commonly identified using culture-dependent methods [37,38], but were also found by Delavenne et al. (2011) in raw milk samples from three different dairy species with D-HPLC, after PCR amplification of the internal transcribed spacer one region [39]. The work of Doyle et al. (2017) showed that the alpha diversity of the milk microbiota from grazing cows was significantly higher in milks from cows receiving a premilking treatment compared to those with no premilking treatment [1]. In contrast to this work, our study is interested in post-milking treatments, but shows no impact of the treatment on the alpha diversity in milk. Consistent with previous studies [8,31], ripening bacteria, Gram-negative bacteria, and, to a lesser extent, lactic acid bacteria were the dominant cultivable populations on the teat surface. Their levels varied according to milking hygiene practices and were slightly higher on the teats of group C cows, confirming the results of Hohmann et al. (2020) on the bacterial load-decreasing effect of post-milking product application [40]. With nearly 800 ASVs, the teat skin has also revealed a high taxonomic diversity, which, in contrast to the milk, was influenced by the post-milking treatment. *Pseudomonas*, a psychrophilic bacteria, and *Staphylococcus*, a potentially pathogenic bacteria, were the two most abundant bacterial genera on the teat skin of cows receiving no post-milking product. This seems to indicate the possible advantage of teat dipping after milking. Teats dipped with product A presented a higher abundance in eight fungal ASVs, including *Debaryomyces prosopidis*, which was characterized by Andrews et al. (2019) as the most abundant species in a teat skin swab through an ITS amplicon sequence analysis [41].

To our knowledge, our study is the first to compare teat surface microbiota on an individual basis or in a pool. We observed no differences between the two types of suspensions, either in terms of level or richness of microorganisms or in terms of beta-diversity, regardless of the dipping applied. Thus, an analysis of a pool of teat surface suspensions from at least seven individuals appeared to reflect the level, richness, and diversity of fungal communities present on the teat surface. This would appear to be less evident for bacteria. Indeed, a differential analysis of microbial taxa abundances according to post-dipping treatment revealed a majority of bacterial differences in the individual

samples, and a majority of fungal differences in the pooled samples. In addition, only the analysis of individual teat suspensions revealed the changes in alpha and beta diversity indices over time, probably as a result of the number of samples involved in the statistical analysis (27 pooled vs 189 individual samples). However, the choice of analyzing a pool of teat surfaces is obvious in terms of analysis time and cost. For this purpose, it may be of interest not to mix individual teat suspensions but to extract a suspension from a mixture of wipes (e.g., five by five) and to perform several replicates of this mixture in order to increase the power of the statistical analysis.

Regardless of the treatment considered, a large part of SCC averaged less than 350,000 cells/mL, which was rather low with regard to the limit set by the Council Directive 92/46/EEC of 1992 ($4 \times 10^5$ cells/mL). It tended to rise over time in relation to the evolution of the lactation stage, as previously reported for dairy herds in the summertime [42,43]. Similarly, the levels of lactic acid bacteria, ripening bacteria, yeasts, and moulds in milk have tended to increase over time, in line with various studies that have shown a seasonal effect on the levels of microorganisms in milk, higher in summer than in winter [34,44]. Milk A appeared to be enriched with bacteria throughout the experiment, as shown by the increase in its bacterial richness indices, but without changing the evenness.

**5. Conclusions**

Our study shows that the tested post-milking product A had no negative effect on the condition of the teat skin or on the microbial levels on the surface of the teats or in the associated milks. It appeared to slightly favour lactic acid bacteria and yeasts and to limit the development of undesirable bacteria in cheese processing, such as *Pseudomonas* and *Staphylococcus*. Our study shows that bacterial and fungal diversity profiles on the teat surface change significantly over time, probably in relation to factors intrinsic to the animals such as stage of lactation, or to environmental factors. To outweigh these effects of time, the effect of applying new postdipping products requires their use over a sufficiently long period, at least 10 weeks. Product A is, therefore, a possible alternative to the chlorhexidine-based product to help preserve microbial communities of interest for cheese production.

**Supplementary Materials:** The following supporting information can be downloaded at: https://www.mdpi.com/article/10.3390/dairy3020021/s1, Figure S1: Experimental Design; Figure S2: Multi-Factor Analysis (MFA) according to post-milking group and period; Figure S3: Bacterial (16S) and fungal (ITS) betadiversity of individual and pooled teat suspensions according to the period and all treatments together; Figure S4: Number of the bacterial (a) and fungal (b) ASVs shared between milk, pooled and individual teat suspensions; Table S1: Cow group characteristics; Table S2: *p*-value of the bacterial (16S) and fungal (ITS) betadiversity of the pooled teat suspensions according to the treatment and the period; Table S3: *p*-value of the bacterial (16S) and fungal (ITS) dispersion of individual teat suspensions evaluated through Tukey HSD test.

**Author Contributions:** Conceptualization, A.C., C.C., P.P. and C.D.; methodology, I.V.-M. and M.B.; software, E.R. and S.T.; validation, I.V.-M., C.D., E.R. and S.T.; formal analysis, I.V.-M.; investigation, M.B. and I.V.-M.; resources, M.B. and I.V.-M.; data curation, E.R., S.T. and I.V.-M.; writing—original draft preparation, I.V.-M. and C.D.; writing—review and editing, I.V.-M., C.D. and C.C.; visualization, I.V.-M. and C.D.; supervision, C.D. and C.C.; project administration, A.C., P.P. and C.C.; funding acquisition, A.C., P.P. and C.C. All authors have read and agreed to the published version of the manuscript.

**Funding:** This research was partly funded by ACTRADE SARL (Beaupréau, France) under the research contract Université Clermont Auvergne n° DRV/VALO 2017-160.

**Institutional Review Board Statement:** Herbipole Marcenat is approved by the French government (agreement n. D15-114-01) to run experiments on living animals. All animal-related procedures were carried out in accordance with the guidelines for animal research of the French Ministry of Agriculture and all other applicable national and European guidelines and regulations for experimentation with animals.

**Informed Consent Statement:** Not applicable.

**Data Availability Statement:** Raw sequence data were deposited at the European Nucleotide Archive of the European Bioinformatics Institute under the BioProject number PRJEB51233.

**Acknowledgments:** The authors would like to thank the staff of the INRAE UE1414 Herbipôle (Marcenat, France) for animal care and technical assistance and Béatrice Desserre (INRAE UMRF, Aurillac, France) for microbiological analyses. They are grateful to Frédérique Chaucheyras-Durand (Lallemand SAS, France) for reviewing.

**Conflicts of Interest:** Actrade provided the tested post-dip product and made a financial contribution to the study. As head of Actrade, Agnès Corbin validated the experimental design of the study, the results and the manuscript. Since then, this activity has been taken over by the Lallemand company for which Agnès Corbin currently works.

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
