# Peer review of "Influence of Post-Milking Treatment on Microbial Diversity on the Cow Teat Skin and in Milk"

_2624-862X, doi:10.3390/dairy3020021_

Round 1

Reviewer 1 Report

General comments:

This is an interesting study to evaluate the effect of two milk products (post-dipping) in the milk and teat microbiota. Overall, no major issues were found.  Nevertheless, some specific points should be addressed to improves the manuscript.

Specific comments:

L79 (Experimental Design): you don’t report if the pre-dipping is or not performed and the products used. Also some description of the cows/groups is welcome (interval calving first samples; mean parity) to characterize the homogeneity of the three groups.  

L157-154 (including Table 1): there are some doubt/apparent inconsistence between text and Table 1; please clarify. Can you aad ±SEM? Also can add P value for each column (e.g., superscript for column and a new right column for lines)? Thanks.

L92: The mixed milk in each group was taken from composite milk at equal proportion, right?

L165-167- I think that you need to report the SCC threshold used to detect SCM. Probably, in this section, will fit better if you remove these 3 cases: the study is about microbiome and not milk pathogenic.

L169-170 (including Fig. 1): slightly lower? We don’t know. You need to build  a statistical model to detect differences between groups and even between periods to known the pattern of each one.

L295: You mean clinical mastitis? There are 3 SCM (two occurrence in groups A and C) and we don’t can ensure your statement contrarily to hyperkeratosis (zero occurrences).

L298: Group C shows 2.15 score at the end of the experiment. Maybe attributing C to group control can avoid mis-interpretations.

L304: Yes, but both are similar to group N (group control), ie, in strict sense both groups don’t differ from N. Probably with more samples, you can reach this significance. So, the term “contradiction” is not entirely correct. Probably the characteristics of both products can explain the differences between them.  

L308: You mean “dairy derivate products”?

Author Response

Comments and Suggestions for Authors

General comments:This is an interesting study to evaluate the effect of two milk products (post-dipping) in the milk and teat microbiota. Overall, no major issues were found.  Nevertheless, some specific points should be addressed to improves the manuscript.

Thank you for your interest in our work

Specific comments:

Point 1: L79 (Experimental Design): you don’t report if the pre-dipping is or not performed and the products used. Also some description of the cows/groups is welcome (interval calving first samples; mean parity) to characterize the homogeneity of the three groups.

Reponse 1: No pre-dipping was performed in this study and it is now precised in the experimental design. Mean values of the main indicators have been added in the Material and Methods section and detailed values are reported in a new supplementary Table S1.

Point 2: L157-154 (including Table 1): there are some doubt/apparent inconsistence between text and Table 1; please clarify. Can you aad ±SEM? Also can add P value for each column (e.g., superscript for column and a new right column for lines)? Thanks.

Reponse 2: The text has been modified to be in adequation with the table in which were added the SEM and the P-value.

Point 3: L92: The mixed milk in each group was taken from composite milk at equal proportion, right?

Response 3: The mixed milk for each group is the total milk of the all animals in each group. This has been specified in the text.

Point 4: L165-167- I think that you need to report the SCC threshold used to detect SCM. Probably, in this section, will fit better if you remove these 3 cases: the study is about microbiome and not milk pathogenic.

Response 4: As suggested, the SCC threshold has been added in the Material and Methods section. No clinical mastitis and only sporadic sub-clinical mastitis (one in each group) were observed.

Point 5: L169-170 (including Fig. 1): slightly lower? We don’t know. You need to build  a statistical model to detect differences between groups and even between periods to known the pattern of each one.

Response 5: We agree with this remark. Figure 1 has been replaced by Table 2 summarising the Anova results on somatic cell counts over time according to the group.

Point 6: L295: You mean clinical mastitis?. There are 3 SCM (two occurrence in groups A and C) and we don’t can ensure your statement contrarily to hyperkeratosis (zero occurrences).

Response 6: The manuscript has been modified to clarify this point.

Point 7: L298: Group C shows 2.15 score at the end of the experiment. Maybe attributing C to group control can avoid mis-interpretations.

Response 7: We agree with the fact that there is a risk of mis-interpretation so the initial C group receiving chlorhexidine-gluconate base has been renamed G (like gluconate) and the initial N (=no dipping) group became C (= control). The modification was made to the whole manuscript as well as to the supplementary data.

Point 8: L304: Yes, but both are similar to group N (group control), ie, in strict sense both groups don’t differ from N. Probably with more samples, you can reach this significance. So, the term “contradiction” is not entirely correct. Probably the characteristics of both products can explain the differences between them.

Response 8: The sentence has been modified to be clearer: indeed another study didn’t highlight any differences in the microbial counts of the milk but it was a iodine post-milking product compared to no post-dipping.

Point 9: L308: You mean “dairy derivate products”?

Response 9: We mean and precise in the text “milk-based fermented products”.

Reviewer 2 Report

Review

for the journal “DAIRY

Article “Influence of post-milking treatment on microbial diversity on 2 the cow teat skin and in milk”

Authors:  Isabelle Verdier-Metz, Céline Delbès , Matthieu Bouchon, Philippe Pradel, Sébastien Theil, Etienne Rifa, Agnès Corbin  and Christophe Chassard

  1. Most of the research on post-milking teat disinfectant are focused on their effects on the bacteria responsible for mastitis. In this study, the authors investigated the effect of various post-milking teat dips on milk and teat microbiota in general. From a methodological point of view, the work is quite detailed and interesting because the high-throughput sequencing of marker genes was applied to characterize the composition of bacterial and fungal communities on the teat skin and in the milk of dairy cows. The authors also evaluated the inter-individual variations in microbial diversity profiles of teat skin over time and in relation to the appearance of the teat-end.
  2. Lines 42-53. There is no literary source "7" in the text?
  3. Line 85. In my opinion, the composition of each group is not homogeneous (6 multiparous and 1 primiparous cows per group). What was the average lactation of each group of cows?
  4. Line 194. The applied statistical analysis methods are not described in the methodology.
  5. Lines 364-370. Conclusions. In my opinion, the practical value of this work is not sufficiently highlighted.
  6. The article is interesting, but the adjustments mentioned are recommended.

Sincerely, reviewer.

Author Response

Comments and Suggestions for Authors

Point 1 : Most of the research on post-milking teat disinfectant are focused on their effects on the bacteria responsible for mastitis. In this study, the authors investigated the effect of various post-milking teat dips on milk and teat microbiota in general. From a methodological point of view, the work is quite detailed and interesting because the high-throughput sequencing of marker genes was applied to characterize the composition of bacterial and fungal communities on the teat skin and in the milk of dairy cows. The authors also evaluated the inter-individual variations in microbial diversity profiles of teat skin over time and in relation to the appearance of the teat-end.

Reponse 1 : Thank you for your interest in our work.

Point 2: Lines 42-53. There is no literary source "7" in the text?

Reponse 2: It has been added.

Point 3: Line 85. In my opinion, the composition of each group is not homogeneous (6 multiparous and 1 primiparous cows per group). What was the average lactation of each group of cows?

Reponse 3: Mean values of the main indicators have been added in the Material and Methods section and detailed values are reported in a new supplementary Table S1.

Point 4: Line 194. The applied statistical analysis methods are not described in the methodology.

Reponse 4: The modification has been done.

Point 5: Lines 364-370. Conclusions. In my opinion, the practical value of this work is not sufficiently highlighted.

Reponse 5: The conclusion has been completed in order to better highlight the interest of our study from a practical point of view, particularly in the context of raw milk cheese production.

Point 6: The article is interesting, but the adjustments mentioned are recommended.

Reponse 6: We have made changes and clarifications where necessary in accordance with the recommendations.

Round 2

Reviewer 1 Report

Dear authors,

Thanks for submitting the revised version, where all comments were addressed.

Please confirm " 350,000 cells/mL" in L175 (the reported SCM threshold for SCC was 300,000 cells/mL) during Proofs.